# Applications of Abscisic Acid and Increasing Concentrations of Calcium Affect the Partitioning of Mineral Nutrients between Tomato Leaf and Fruit Tissue

**T. Casey Barickman** [1,*] **, Dean A. Kopsell** [2] **and Carl E. Sams** [3]

1    Mississippi State University, North Mississippi Research and Extension Center, Verona, MS 38879, USA
2    Department of Environmental Horticulture, the University of Florida, Gainesville, FL 32611, USA
3    Department of Plant Sciences, the University of Tennessee, Knoxville, TN 37966, USA
*    Correspondence: t.c.barickman@msstate.edu; Tel.: +1-662-566-2201

**Abstract:** This study examined how abscisic acid (ABA) and calcium (Ca) concentrations in nutrient solution affect concentrations of mineral nutrients in tomato leaves and fruit. Tomato plants were grown in a greenhouse at 25/20 °C (day/night) under a 16 h photoperiod. Plants were treated with different concentrations of ABA and Ca. Calcium was applied via the irrigation lines at 60, 90, or 180 mg·L$^{-1}$. ABA was applied as a combination of foliar sprays and root applications. For foliar ABA applications, treatments consisted of deionized (DI) water control (0.0 mg·L$^{-1}$ ABA) or 500 mg·L$^{-1}$ ABA. For ABA root applications, treatments consisted of no ABA control (0.0 mg·L$^{-1}$ ABA) or 50 mg·L$^{-1}$ ABA applied via the irrigation lines. Results indicate that mineral nutrient concentrations in tomato leaf and fruit tissue varied in connection with each exogenous application of ABA. Variability in mineral nutrient concentration depended on if ABA was applied to the leaf or root tissue. Additionally, increasing Ca treatment concentrations either decreased or did not change mineral nutrients in tomato and fruit tissue. Thus, tomato plants react to acquiring mineral nutrients in numerous mechanisms and, depending on how the applications of exogenous ABA are applied, can have varying effects on these mechanisms.

**Keywords:** Potassium; magnesium; iron; boron; abiotic stress

## 1. Introduction

Mineral nutrients are essential to cell life and are of paramount importance to assess how plants control and regulate their nutritional status [1]. Both macronutrient and micronutrient mineral elements are acquired through the root system as ions from the rhizosphere solution [2], with a balanced supply of these essential elements needed for optimal plant growth and development. Thus, plants regulate mechanisms that govern ion acquisition, accumulation, and homeostasis [3]. The availability and root uptake, translocation, and allocation of nutrients are affected by nutrient source and other physical, chemical, and biological characteristics caused by environmental factors, such as pH, temperature, and the concentrations and interactions of endogenous plant hormones.

Environmental stresses frequently influence vegetative development by altering the homeostasis and distribution of mineral nutrients within plant tissue [4]. Altering the homeostasis and distribution of mineral nutrients such as Ca and B can cause physiological disorders, such as blossom-end rot in tomato and pepper fruit [5,6], hollow heart in the stems of broccoli [7], and bitter pits in apples [8]. Previous research has demonstrated that the plant hormone abscisic acid (ABA) helps plants acclimate to environmental stresses such as drought, extreme temperatures, and excess light [9,10]. The impact of

environmental stress, and indirectly ABA, influences nutritional fluxes in plant tissues such as leaves, roots, and fruit. For example, under high temperature stress, ABA can indirectly enhance potassium (K) absorption in cucumber (*Cucumis sativus*) [11], and it can promote calcium (Ca) uptake and distribution in tomato fruit [12–14]. Furthermore, ABA can inhibit K channel function to ensure reduced K loading into the xylem, a mechanism that helps maintain root turgor during drought conditions [15]. Thus, ABA can indirectly regulate mineral nutrient homeostasis and distribution in various plant tissues.

Calcium ions can influence the selectivity of ion uptake, and the relative accumulation of mineral nutrients in the plant root rhizosphere. For example, under Ca limiting conditions, there are differences in the K–sodium (Na) uptake into the root tissue. However, when Ca level is sufficient, the uptake of K is favored at the expense of Na [2]. These shifts in the K–Na uptake ratio are likely because extracellular Ca inhibits Na influx through voltage-sensitive cation channels [16–18]. The uptake of cations and anions occurs through different transport proteins. The uptake of one nutrient can influence the uptake of another indirectly through effects on membrane potential, electrochemical gradients, or from feedback regulation through plant growth and metabolism [2]. Thus, when one nutrient is supplied in elevated concentrations, it can influence the uptake of another nutrient. Calcium, in contrast to other macronutrients, is located in cell walls at relatively high concentrations. To avoid toxicity and precipitation, Ca is highly regulated in the cell cytoplasm. This unique distribution is mainly the result of a large number of binding sites in the cell wall and the act of being a signaling molecule in the cytoplasm [19].

Abscisic acid has also been shown to be a signaling molecule in plant tissue. Evidence has indicated that, in drought conditions, ABA acts as a non-hydraulic chemical signal from the roots to the shoot. This signal decreases stomatal conductance and reduces transpiration, gas exchange [20], and, ultimately, plant growth. Additionally, ABA in the xylem sap is also affected by nitrogen (N), phosphorus (P), and K concentrations [21–23]. The root-derived hormonal signals in the xylem sap can also affect the long-distance transport of nutrients to the shoots. For instance, these signals can affect the volume flow rate in the xylem, rate of xylem-phloem transfer, and nutrient distribution within the shoot [2].

This study aims to examine the interactive effects of exogenous applications of ABA and increasing Ca concentrations on mineral nutrient concentrations of tomato leaf and fruit tissue. ABA can be applied to the plant either via the root or as a spray to the vegetative tissue. When applied to the root tissue, research has demonstrated that ABA improves water-use efficiency by closing the stomata and affecting plant growth by reducing transpiration through xylem tissue to the desired tissue [24]. Research on exogenous applications of ABA and its effects on mineral nutrient uptake and distribution has mainly focused on Ca. For example, recent research has demonstrated that foliar and root applications are effective at increasing Ca in the fruit tissue and decreasing the incidence of blossom-end rot [14].

Nevertheless, there is minimal information on how exogenous applications of ABA impact the uptake and distribution of other mineral nutrients, such as K, iron (Fe), and molybdenum (Mo), in tomato leaf and fruit tissue. Therefore, the purpose of this study was to examine how ABA applications to the leaf and root tissue, individually and in combination, and increasing concentrations of Ca in the nutrient solution affect the partitioning and distribution of mineral nutrients between the leaves and fruit of tomato plants. The hypothesis was that ABA applications would decrease mineral nutrients in the leaf tissue and increase concentrations, especially in the distal partition of the fruit tissue. This study demonstrated that ABA could regulate the partitioning of mineral nutrients from vegetative tissue to the fruit tissue, thus increasing the nutritional value.

## 2. Materials and Methods

### 2.1. Plant Culture and Harvest

The experimental design, treatment applications, and tomato leaf and fruit harvest followed the procedures of Barickman et al. [14]. Concisely, seeds of "Mountain Fresh Plus" tomato (Johnny's Selected Seed, Waterville, ME, USA) were sown into Pro-Mix BX soilless medium (Premier Tech Horticulture, Québec, QC, Canada) and germinated in greenhouse conditions (Knoxville, TN, USA; 35°N Lat.) at 25/20 °C (day/night). Natural photoperiod and intensity of sunlight for tomato production in the greenhouse were supplemented with 24 individual 1000 W high-pressure sodium lights under a 16 h photoperiod. The lights delivered an average of 900 $\mu$mol·m$^{-2}$·s$^{-1}$ over the entire photoperiod. Light intensity readings were taken 1.22 m off the ground. At 30 d after seeding, the plantlets were transferred to 11 L Dutch pots (Tek Supply, Dyersville, IA, USA) filled with Sunshine® Pro Soil Conditioner (Sungro Horticulture, Agawam, MA, USA). Tomato plants were fertigated four times daily for 10 min each irrigation cycle. Elemental concentrations of the nutrient solutions were (mg·L$^{-1}$): N (180), P (93.0), K (203.3), Mg (48.6), sulfur (S; 96.3), Fe (1.0), boron (B; 0.25), Mn (0.25), zinc (Zn; 0.025), copper (Cu; 0.01) and Mo (0.005). There were two identical experiments conducted. The first experiment was conducted in fall 2012 and the second in spring 2013. The experimental design was a randomized complete block with a 3 × 4 factorial arrangement of treatments that consisted of six blocks and two replications of each treatment per block, with individual plants representing an experimental unit. Treatments of Ca were applied via the irrigation lines at 60, 90, or 180 mg·L$^{-1}$ Ca, given as calcium nitrate. The ABA (VBC-30151; Valent BioSciences, Libertyville, IL, USA) treatments were applied as foliar sprays, root applications, or the combination of foliar and root application. For foliar ABA applications, treatments consisted of DI water control (0.0 mg·L$^{-1}$ ABA) or 500 mg·L$^{-1}$ ABA. The ABA root treatment applications consisted of a control (0.0 mg·L$^{-1}$ ABA) or 50 mg ABA·L$^{-1}$ applied via the irrigation lines. The combination treatment consisted of a foliar ABA spray of 500 mg·L$^{-1}$ ABA and a root application of 50 mg·L$^{-1}$ ABA. The ABA spray treatments were applied once weekly from anthesis to final harvest. Fruit tissues were harvested 84 d after seeding when tomato fruit were red ripe. Tomato fruit samples were taken from the second cluster for mineral nutrient analysis. Subsequently, fruits were sorted using the USDA tomato color for fruit [25] and size classification into extra-large, large, medium, and small [26]. Tomato fruit with blossom-end rot (BER) were categorized separately. Harvested fruit samples were stored at −80 °C before analysis. Leaf samples for each replication were taken from the first leaf about the second cluster on the final harvest of fruit from that cluster for mineral nutrient analysis.

### 2.2. Elemental Nutrient Determination

Nutrient analyses were conducted according to Barickman et al. [27]. Briefly, sample analysis was performed using a 5.0 g subsample of fresh fruit tissue, which was combined with 10 mL of 70% HNO$_3$ and digested in a microwave digestion unit (model: Ethos, Milestone Inc., Shelton, CT, USA). Leaves were dried for 48 h in a forced air oven (model large; Fisher Scientific, Atlanta, GA, USA) at 65 °C. Nutrient analyses were conducted using an inductively coupled plasma mass spectrometer (ICP-MS; Agilent Technologies, Inc., Wilmington, DE, USA). Tissue nutrient concentrations are expressed on a dry weight (DW) basis.

### 2.3. Statistical Analysis

Results from the two separate experiments were statistically similar. Therefore, data were pooled and analyzed together for treatment means. Statistical analyses of data were performed using SAS (version 9.4; SAS Institute, Cary, NC, USA). Data were analyzed using the PROC GLIMMIXED analysis of variance followed by mean separation. Duncan's multiple range test ($P \leq 0.05$) was used to differentiate between ABA and Ca application classifications.

## 3. Results

The application of ABA significantly affected concentrations of mineral nutrients in tomato leaf tissue (Table 1). The most dramatic change in nutrient concentrations was for Mg in the leaf tissue when ABA was applied to both leaf and root tissue. There were also significant differences in P and K when ABA was applied as a foliar spray treatment with a 15% and 27% decrease in concentration when compared to the control treatment, respectively. Interestingly, P was increased by 16% in tomato leaf tissue when ABA was applied as a foliar spray treatment. There were also significant differences in micronutrient concentrations in tomato leaf tissue (Table 1). The micronutrients B, Zn, and Cu decreased the most when ABA was applied to both the leaf and root tissue. There was a 37%, 39%, and 35% decrease in B, Zn, and Cu in the leaf tissue when comparing the combination of spray and root tissue ABA treatment to the control, respectively.

**Table 1.** Mineral nutrients in leaf tissue of 'Mt. Fresh Plus' tomato plants grown in a greenhouse and treated with DI water, as control, or exogenous applications of abscisic acid (ABA); either foliar spray (500 mg·L$^{-1}$), root drench (50 mg·L$^{-1}$), or a combination of foliar spray and drench. Leaf tissue for mineral nutrient analysis was taken above the second cluster from each treatment and replication at 84 d after seeding.

| ABA Treatment | Leaf Tissue Mineral Elements (mg·g$^{-1}$ Dry Weight) [a] | | | | | | | | | |
|---|---|---|---|---|---|---|---|---|---|---|
| | Mg | P | S | K | Mn | B | Fe | Zn | Cu | Mo [b] |
| Control | 11.32 | 8.40 | 17.70 | 48.99 | 0.37 | 0.19 | 0.15 | 0.031 | 0.029 | 2.20 |
| 500 mg·L$^{-1}$ Spray | 11.28 | 7.12 | 16.04 | 35.81 | 0.34 | 0.13 | 0.13 | 0.022 | 0.020 | 2.20 |
| 50 mg·L$^{-1}$ Root | 8.66 | 9.97 | 16.40 | 45.55 | 0.38 | 0.17 | 0.13 | 0.022 | 0.023 | 3.40 |
| 500 mg·L$^{-1}$ Spray + 50 mg·L$^{-1}$ Root | 8.12 | 8.09 | 16.16 | 37.96 | 0.34 | 0.12 | 0.12 | 0.019 | 0.019 | 3.40 |
| *P*-Value [c] | *** | *** | ns | *** | ns | *** | ns | *** | *** | *** |

[a] The standard error of the mean was Mg ± 0.49; P ± 0.41; S ± 0.59; K ± 2.01; Mn ± 0.02; B ± 0.01; Fe ± 0.01; Zn ± 0.005; Cu ± 0.001; Mo ± 0.0002. [b] Mo given as μg·g$^{-1}$ dry weight. [c] ns, *** indicate non-significant or significant at $P \leq 0.001$, respectively.

Abscisic acid had a significant effect on the micronutrient concentrations of B, Mn, and Mo in tomato fruit tissue (Table 2). There were decreases in B concentrations when ABA was applied in general. For example, when ABA was foliar applied to tomato leaf tissue, leaf tissue B decreased by 12% when compared to the control treatment. Similarly, when ABA was applied to both the leaf and root tissue of tomato plants, there was a 17% decrease in B concentration in the fruit tissue compared to the control treatment. Both Mn and Mo increased in tomato fruit tissue by 15% and 25% when ABA was applied as a foliar spray, respectively, compared to the control treatment. Furthermore, there were significant changes in macronutrients when ABA was applied to tomato plants (Table 2). The K and P concentrations in the fruit tissue were decreased by 9% and 11%, respectively, compared to the control treatment when ABA was applied as a foliar spray. However, there were slight increases in concentrations of K and P in the fruit tissue by 3% and 4%, respectively, when ABA was applied to tomato root through drip irrigation compared to the control treatment.

**Table 2.** Mineral nutrients in fruit tissue of 'Mt. Fresh Plus' tomato plants grown in a greenhouse and treated with exogenous applications of abscisic acid (ABA). Fruit tissue was harvested at 84 d after seeding from the second cluster at the red ripe stage for each treatment and replication for mineral nutrient analysis.

| ABA Treatment | Fruit Tissue Mineral Elements [a] | | | | | | | | | |
| --- | --- | --- | --- | --- | --- | --- | --- | --- | --- | --- |
| | (mg·g$^{-1}$ Dry Weight) | | | | | | (µg·g$^{-1}$ Dry Weight) | | | |
| | **Mg** | **P** | **S** | **K** | **Mn** | **B** | **Fe** | **Zn** | **Cu** | **Mo** |
| Control | 6.37 | 18.55 | 6.46 | 160.27 | 75.00 | 57.50 | 258.33 | 239.17 | 118.33 | 2.50 |
| 500 mg·L$^{-1}$ Spray | 5.97 | 16.45 | 6.73 | 146.69 | 88.33 | 50.83 | 274.17 | 223.33 | 132.50 | 3.33 |
| 50 mg·L$^{-1}$ Root | 6.63 | 19.39 | 7.94 | 166.57 | 85.83 | 54.17 | 299.17 | 215.83 | 121.67 | 3.33 |
| 500 mg·L$^{-1}$ Spray + 50 mg·L$^{-1}$ Root | 6.05 | 16.27 | 6.03 | 144.83 | 87.50 | 47.50 | 237.50 | 189.17 | 118.33 | 3.33 |
| *P*-Value [b] | ns | ** | ns | * | ** | ** | ns | ns | ns | *** |

[a] The standard error of the mean was Mg ± 0.50; P ± 1.49; S ± 1.00; K ± 13.26; Mn ± 6.67; B ± 5.00; Fe ± 30.83; Zn ± 38.33; Cu ± 29.17; Mo ± 0.67. [b] ns, *, **, *** indicate non-significant or significant at $P \leq 0.05$, 0.01, and 0.001, respectively.

In the current study, partitioning coefficients were calculated to determine the partitioning of mineral nutrients in tomato leaves and fruit tissue. Partitioning coefficients were calculated, with values greater than one indicating a higher concentration of mineral nutrients in the leaf compared to the fruit tissue. The mineral nutrients B, Mg, and Mn exhibited higher concentrations in the leaf compared to the fruit tissue. On the other hand, coefficients that were calculated with values less than one indicated a lower concentration of mineral nutrients in the leaf compared to the fruit tissue. The mineral nutrients P, K, Fe, Zn, and Cu were at higher concentrations in the fruit compared to the leaf tissue. In general, foliar applications of ABA had a significant effect on the partitioning of mineral nutrients of the leaf and fruit tissue. For example, ABA significantly increased B, Mg, K, Mn, Cu, and Zn in fruit tissue rather than leaf tissue when compared to the non-treated control (Table 3). Specifically, fruit tissue K had a lower partitioning coefficient when ABA was applied as a foliar treatment, and there was an increase in the ratio by 22% when compared to the control treatment. Furthermore, fruit tissue Mg had a lower partitioning coefficient when ABA was applied as a foliar treatment, and there was an increase in the ratio by 26% when compared to the control treatment.

**Table 3.** Mineral nutrient partitioning coefficient for leaf to fruit ratio of nutrients in 'Mt. Fresh Plus' tomato plants grown in a greenhouse and treated with exogenous applications of abscisic acid (ABA).

| ABA Treatment | Partitioning Coefficient (Leaf Nutrient/Fruit Nutrient) [a] | | | | | | | | | |
| --- | --- | --- | --- | --- | --- | --- | --- | --- | --- | --- |
| | **Mg** | **P** | **S** | **K** | **Mn** | **B** | **Fe** | **Zn** | **Cu** | **Mo** |
| Control | 1.90 | 0.47 | 3.54 | 0.32 | 5.26 | 3.56 | 0.66 | 0.15 | 0.48 | 1.36 |
| 500 mg·L$^{-1}$ Spray | 1.41 | 0.45 | 3.03 | 0.25 | 3.99 | 2.69 | 0.67 | 0.12 | 0.34 | 1.09 |
| 50 mg·L$^{-1}$ Root | 1.87 | 0.54 | 3.40 | 0.30 | 4.88 | 3.71 | 0.57 | 0.12 | 0.41 | 1.19 |
| 500 mg·L$^{-1}$ Spray + 50 mg·L$^{-1}$ Root | 1.50 | 0.53 | 3.81 | 0.27 | 4.20 | 2.73 | 0.69 | 0.13 | 0.36 | 0.54 |
| *P*-Value [b] | *** | ** | ns | ** | ** | *** | ns | * | * | ns |

[a] The standard error of the mean was Mg ± 0.12; P ± 0.03; S ± 2.63; K ± 0.01; Mn ± 0.32; B ± 0.25; Fe ± 0.07; Zn ± 0.02; Cu ± 0.06; Mo ± 0.53. [b] ns, *, **, *** indicate non-significant or significant at $P \leq 0.05$, 0.01, and 0.001, respectively.

Tomato fruit was separated into proximal and distal ends and analyzed for mineral nutrient content after being treated with ABA. The micronutrients B, Fe, and Mo were significantly affected in the distal fruit tissue (Table 4). For example, B, Fe, and Mo showed 24%, 38%, and 25% increases, respectively, in concentrations in the distal compared to the proximal fruit tissue. Furthermore, there were significant changes in macronutrient concentrations for the tomato fruit tissue (Table 4). There were 7% and 17% increases in P and Mg concentrations in the distal fruit tissue when compared to the proximal fruit tissue, respectively.

**Table 4.** Mineral nutrients in fruit tissue of 'Mt. Fresh Plus' tomato plants grown in a greenhouse and treated with exogenous applications of abscisic acid (ABA). Fruit tissue was harvested at 84 d after seeding from the second cluster at the red ripe stage for each treatment and replication for mineral nutrient analysis.

| Location | Fruit Tissue Mineral Elements [a] | | | | | | | | | |
| | (mg·g$^{-1}$ Dry Weight) | | | | | (µg·g$^{-1}$ Dry Weight) | | | | |
| | Mg | P | S | K | Mn | B | Fe | Zn | Cu | Mo |
|---|---|---|---|---|---|---|---|---|---|---|
| Proximal | 5.68 | 17.06 | 6.77 | 157.62 | 83.33 | 45.00 | 204.17 | 215.83 | 129.17 | 2.50 |
| Distal | 6.83 | 18.27 | 6.81 | 151.55 | 85.00 | 59.17 | 330.00 | 217.50 | 117.50 | 3.33 |
| *P*-Value [b] | *** | * | ns | ns | ns | *** | *** | ns | ns | ** |

[a] The standard error of the mean was Mg ± 0.46; P ± 1.41; S ± 0.88; K ± 12.56; Mn ± 6.67; B ± 5.00; Fe ± 28.33; Zn ± 37.50; Cu ± 25.83; Mo ± 0.67.　[b] ns, *, **, *** indicate non-significant or significant at $P \leq 0.05$, 0.01, and 0.001, respectively.

There were significant linear trends for the distribution of mineral nutrients in tomato leaf tissue when plants were treated with increasing concentrations of Ca (Table 5). The macronutrients Mg, P, and K decreased in concentration with increasing Ca treatments. For instance, Mg, P, and K decreased 30%, 14%, and 11%, respectively, in tomato leaf tissue when Ca treatments increased from 60 to 180 mg·L$^{-1}$, respectively. There were no significant changes for S in the leaf tissue when Ca treatments increased from 60 to 180 mg·L$^{-1}$. Micronutrient concentrations decreased in tomato leaf tissue with increasing Ca treatments. For example, B, Mn, and Fe decreased 13%, 16%, and 31% in the leaf tissue when Ca treatments increased from 60 to 180 mg·L$^{-1}$, respectively. In addition, Zn, Mo, and Cu decreased 12%, 17%, and 20% in tomato leaf tissue when Ca treatments increased from 60 to 180 mg·L$^{-1}$, respectively.

**Table 5.** Mineral nutrients in leaf tissue of 'Mt. Fresh Plus' tomato plants grown in a greenhouse and treated with increasing concentrations of calcium, given as calcium nitrate. Leaf tissue for mineral nutrient analysis was taken above the second cluster for each treatment and replication at 84 d after seeding.

| Calcium Treatments | Leaf Tissue Mineral Elements (mg·g$^{-1}$ Dry Weight) [a] | | | | | | | | | |
| | Mg | P | S | K | Mn | B | Fe | Zn | Cu | Mo |
|---|---|---|---|---|---|---|---|---|---|---|
| 60 | 11.09 | 8.61 | 16.22 | 43.12 | 0.38 | 0.16 | 0.16 | 0.025 | 0.024 | 0.0030 |
| 90 | 10.72 | 8.93 | 16.96 | 45.07 | 0.38 | 0.16 | 0.14 | 0.024 | 0.024 | 0.0030 |
| 180 | 7.72 | 7.42 | 16.55 | 38.05 | 0.32 | 0.14 | 0.11 | 0.022 | 0.020 | 0.0024 |
| *P*-Value [c] | *** | ** | ns | ** | ** | * | *** | ns | ** | ** |
| Contrast | | | | | | | | | | |
| Linear | *** | ** | ns | ** | ** | ** | *** | * | ** | ** |
| Quadratic | ns | ns | ns | ns | ns | ns | ns | ns | ns | ns |

[a] The standard error of the mean was Mg ± 0.43; P ± 0.35; S ± 0.52; K ± 1.84; Mn ± 0.02; B ± 0.01; Fe ± 0.01; Zn ± 0.002; Cu ± 0.001; Mo ± 0.0002.　[b] ns, *, **, *** indicate non-significant or significant at $P \leq 0.05$, 0.01, and 0.001, respectively.

Calcium did not significantly affect mineral nutrient concentrations in tomato fruit tissue (Table 6). However, there were linear and quadratic trends in the data with increasing concentrations with Ca treatments from 60 to 180 mg·L$^{-1}$. For instance, Mn showed a negative linear trend, decreasing 9%, when Ca treatments increased from 60 to 180 mg·L$^{-1}$. On the other hand, K showed a positive linear trend, increasing 9%, when Ca treatments increased from 60 to 180 mg·L$^{-1}$ (Table 6). There were positive quadratic trends for Mo and Mg when Ca treatments increased from 60 to 180 mg·L$^{-1}$. For example, Mg demonstrated an increase of 7% when the Ca treatment increased from 60 to 90 mg·L$^{-1}$, then decreased in concentration by 9% when the Ca treatment increased from 90 to 180 mg·L$^{-1}$ (Table 6).

**Table 6.** Mineral nutrients in fruit tissue of 'Mt. Fresh Plus' tomato plants grown in a greenhouse and treated with increasing concentrations of calcium, given as calcium nitrate. Fruit tissue was harvested at 84 d after seeding from the second cluster at the red ripe stage for each treatment and replication for mineral nutrient analysis.

| Calcium Treatments | Fruit Tissue Mineral Elements [a] | | | | | | | | | |
| | (mg·g$^{-1}$ Dry Weight) | | | | | (µg·g$^{-1}$ Dry Weight) | | | | |
| | Mg | P | S | K | Mn | B | Fe | Zn | Cu | Mo |
| 60 | 6.14 | 17.07 | 6.68 | 146.32 | 87.50 | 50.83 | 260.83 | 221.67 | 119.17 | 3.33 |
| 90 | 6.59 | 17.80 | 7.10 | 156.55 | 85.83 | 53.33 | 274.17 | 214.17 | 119.17 | 3.33 |
| 180 | 6.03 | 18.12 | 6.59 | 160.90 | 80.00 | 52.50 | 266.67 | 214.17 | 130.00 | 2.50 |
| *P*-Value [b] | ns | ns | ns | ns | ns | ns | ns | ns | ns | * |
| Contrast | | | | | | | | | | |
| Linear | ns | ns | ns | * | * | ns | ns | ns | ns | ns |
| Quadratic | * | ns | ns | ns | ns | ns | ns | ns | ns | * |

[a] The standard error of the mean was Mg ± 0.48; P ± 1.45; S ± 0.95; K ± 12.94; Mn ± 6.67; B ± 5.00; Fe ± 30.00; Zn ± 37.50; Cu ± 27.50; Mo ± 0.67. [b] ns, *, indicate non-significant or significant at *P* ≤ 0.05, respectively.

## 4. Discussion

Long-distance xylem and phloem transport of mineral nutrients are essential for the growth and development of plant tissues to carry out critical functions such as photosynthesis and reproduction. When plants are exposed to environmental stresses, such as high temperatures and light intensities, drought, and even cases of high and low vapor pressure deficits, photosynthesis can be decreased leading to the slowing of growth and other detrimental metabolic activities. Under these stress conditions, plants synthesize ABA from the 2-C-methyl-D-erythritol-4-phosphaste (MEP) and carotenoid biosynthetic pathway [28,29]. Additionally, ABA can be transported long-distance within the plant via its inactive form of ABA-glucose (ABA-GE) and is released by β-D-glucosidase under stress conditions [30]. Plants under these stress conditions can have issues with mineral nutrient assimilation and distribution. For example, tomato plants under environmental stress conditions can produce fruit with blossom-end rot [31]. However, previous research has indicated that ABA indirectly helps to assimilate and distribute Ca into fruit tissue that have difficulty under environmental stress conditions [12–14]. These findings have significant impacts on how to decrease physiological disorders such as blossom-end rot. Consequently, in the current study, ABA was applied to the foliage and root tissue of tomato plants to determine how the plants assimilate and distribute mineral nutrients other than Ca. For example, if ABA was applied to the foliar tissue of tomato plants, mineral nutrient concentrations decreased or showed no significant change from the control. Applications of ABA to tomato plant roots lowered Mg in the leaf tissues more than ABA applied to the foliar tissue. Applying ABA to tomato plant roots may induce a stress response that decreases the uptake and/or distribution of Mg into the leaf tissue. Previous research indicated that applications of ABA at 100 mg·L$^{-1}$ to sweet pepper (*Capsicum annum*) leaf tissue did not significantly change Mg from the non-treated control [31]. Research has focused on mineral nutrient limitation and salt stress, and how these environmental stress factors affect the biosynthesis of ABA [22,32]. There has been a lack of research on how exogenous application of ABA affects the concentrations of mineral nutrients in leaf tissue. This study demonstrated that ABA has varying impacts on mineral nutrient concentrations in tomato leaf tissue, depending on how it is administered. Thus, the data were pooled and analyzed to compare against the control treatment for how ABA distributed mineral nutrients between the proximal and distal fruit tissue. In general, ABA increased the mineral nutrients in the distal fruit tissue. Previous research has indicated that ABA increased Ca concentrations in the distal end of tomato fruit tissue and decreased the incidence of blossom-end rot [13,14]. This study supports the hypothesis that ABA indirectly regulates mineral nutrient distribution in plant tissues, especially in the fruit.

Previous research has determined partitioning coefficients for plant biomass [33] and carbon fluxes from shoot to root (phloem transport) and that of mineral nutrient from roots to shoots (xylem transport) in various plant species [34]. Fujita et al. [35] demonstrated that fruit growth, cell expansion, and the partitioning of carbon, N, and P to the fruit were significantly reduced in tomato under P deficiency. In instances of plant stress, such as nutrient deficiency or temperature stress, ABA acts as a signal that regulates transpiration water loss and gas exchange [36], and Ca partitioning and distribution [12–14]. Therefore, such as in the current study, ABA could trigger mechanisms that partition and distribute mineral nutrients in plant tissues. Plants respond to adverse environmental conditions through the synthesis of ABA [37]. Because mineral nutrients are taken up by the roots, the signal pathway for ABA because of nutrient stress would be the same as the pathway of nutrients to the shoot [38]. When ABA is exogenously applied, such as in the current study, similar signals are transduced to mimic adverse environmental conditions, such as nutrient stress. Therefore, ABA can have long-distance impacts on the transport of nutrients via the volume flow rate, the rate of xylem-phloem transfer, and the nutrient distribution within the shoot. The increase of ABA under adverse environmental conditions can cause restriction of nutrient flow from root to shoot, as demonstrated in the current study, and may be due to the reduction of transpiration because of decreased stomatal conductance and reduction of leaf expansion.

Abscisic acid signaling events at the membrane level involve many mineral nutrient protein channels and transporters. One mechanism that has been extensively studied is how ABA indirectly controls stomatal closure. For example, in the event of stomatal closure, ABA activates the efflux of $Cl^-$ through ion channels and in parallel stimulates $K^+$ efflux and inhibits $K^+$ influx ion channels to promote stomatal closure [39]. Abscisic acid also regulates repetitive cytosolic free $Ca^{2+}$ elevation in guard cells [40]. These responses in guard cells have revealed that ABA mediates $Ca^{2+}$ sensors so they can respond to environmental stress events, such as in drought conditions [41,42]. Previous research has demonstrated that net efflux of $K^+$ from the stellar parenchyma cells decreases after treatment with ABA [43–46]. In addition, Roberts [47] used patch clamp techniques on root tissue to demonstrate that ABA reduced $K^+$ channel activity in xylem parenchyma cells. Therefore, the reduction of $K^+$ permeability in the xylem parenchyma plasma membranes is affected by ABA and, as such, reduces $K^+$ movement via the xylem. In the current study, application of ABA treatments decreased K in tomato leaf and fruit tissue. From previous studies, there may be evidence that ABA inhibits the $K^+$ permeability to the xylem and; therefore, reduces its concentrations in the leaf and fruit tissue. However, in ABA root treatments, K increased slightly in the fruit tissue and was significantly similar to concentrations in the leaf tissue compared to the non-treated control.

In recent years, research has demonstrated how ABA influences plant metabolites, such as carotenoids [48,49], flavonoids [50], soluble sugars [49,51], and Ca uptake and distribution [12–14,52,53]. A study conducted by Pérez-Jiménez et al. [31] demonstrated the effect of ABA on mineral nutrient concentrations, soluble sugars, and yield of sweet pepper fruit. Previous research by de Freitas et al. [52,53] and Barickman et al. [12–14,48,51] on how ABA affects Ca concentrations, soluble sugar content, and yield in tomato plants presented opposing conclusions. For example, the work by Pérez-Jiménez et al. [31] did not find differences in Ca and soluble sugar concentrations in sweet pepper fruit tissue. In addition, there were significant increases in Fe concentrations in sweet pepper fruit tissue. The current study demonstrated no significant effects on Fe concentration in tomato fruit tissue when tomato plants were treated with exogenous application of ABA. While the results did not demonstrate a significant increase, there was an increasing trend in the data. A big difference between the studies was that previous research treated tomato plants with five times the amount of ABA used by Pérez-Jiménez et al. [31], who applied foliar ABA at 100 mg·$L^{-1}$ to sweet pepper plants. Furthermore, there may be differences in how plant species respond metabolically to exogenous applications of ABA.

In the current study, Ca treatments affected the concentrations of mineral nutrients in the leaf tissue, except for S and Zn. However, there was a negative linear trend for Zn concentrations in the leaf

tissue. Previous research has demonstrated that Ca influences the acquisition of other cations directly by competing for transport by membrane proteins. Physiochemical interactions between ions can occur at different levels, such as (1) in the uptake of these elements; (2) in their transport through the xylem and phloem; and (3) in their function at different exchange sites [54]. However, previous research found similar results in macronutrients, but differing results for micronutrients. Kopsell et al. [55] found that increasing Ca concentrations in the nutrition solution decreased Mg, P, K, S, and Zn in kale leaf tissue. In addition, they also found that Ca concentrations did not affect B, Cu, Fe, Mn, or Mo concentrations in kale leaf tissue. Further, research has indicated that increasing Ca concentrations in the rhizosphere solution can also reduce the uptake of Mn [56] and Zn [57]. Consequently, Ca may indirectly inhibits cations by competing for binding sites in the rhizosphere and apoplast, inhibiting transport by membrane proteins, or by regulating transport processes via changes in cytosolic Ca levels [58]. Furthermore, Mg functions in P metabolism, plant respiration, and formation of the chlorophyll molecules [2]. The results of the current study indicated that increasing Ca treatment concentrations diminished the concentration of Mg in tomato leaf tissue, which previous research has demonstrated in tobacco (*Nicotiana tabacum*) [59] and kale (*Brassica oleracea* var *Acephala*) [55]. The metabolism of P and its assimilation into molecules such as phospholipids, ATP, and nucleic acids is essential to plant growth and development [60]. The assimilation of K into plant cells is necessary for osmoregulation [61] and enzyme activation [62]. Thus, as previous studies have indicated, increasing Ca fertilization can have a negative effect on mineral nutrient acquisition and distribution. The essential element Ca functions in the regulation of several enzymes and contributes to cell wall structure and reduction of nitrate ($NO_3^-$) in plant tissues [2]. Calcium is taken up passively by plant roots and transported through the xylem with the transpiration stream. It also competes for uptake with ammonium ($NH_4^+$), K, and Mg [63]. How Ca influences uptake of mineral nutrients reflects differences among plant species.

This study examined how exogenous applications of ABA and increasing concentrations of Ca in the nutrient solution affected the concentrations of mineral nutrients in tomato leaf and fruit tissue. Increasing Ca treatments decreased mineral nutrients in the leaf tissue, while the effect on fruit tissue depended on the individual mineral nutrient. For example, there were decreases in Mg, K, Fe, and Cu in the leaf tissue. There were also decreasing linear and quadradic trends for K and Mg, respectively. The results also indicated that ABA influences the concentration of mineral nutrients in leaf and fruit tissue. ABA decreased the concentrations of mineral nutrients such as Mg, B, Fe, Zn, and Cu in the leaf tissue. In other instances, ABA increased concentrations of mineral nutrients such as P, K, Mn, and Mo in the fruit tissue. The combination of foliar and root applications of ABA had the most detrimental effects by decreasing the concentration of mineral nutrients for most elements. Previous research found similar positive trends affected by how ABA was applied, indicating that application of ABA as a foliar spray improved tomato fruit quality by increasing Ca concentrations and soluble sugars in the fruit tissue when compared to applications of ABA to the root tissue. For example, Barickman et al. [13,14] indicated that applying ABA as a foliar treatment significantly increased Ca concentration in the distal tissue of tomato fruit and decreased the incidence of BER compared to the non-treated control and root tissue applications. In addition, Barickman et al. [48,51] demonstrated that foliar ABA increased soluble sugars such as glucose and fructose in tomato fruit tissue. Thus, even though there may have been decreases in mineral nutrient concentrations in tomato leaf tissue, previous research demonstrated that there were no significant reductions in yield [12–14].

**Author Contributions:** Conceptualization, T.C.B. and C.E.S.; methodology, T.C.B., D.A.K., and C.E.S.; validation, T.C.B., D.A.K., and C.E.S.; formal analysis, T.C.B., D.A.K., and C.E.S.; investigation, T.C.B.; resources, D.A.K. and C.E.S.; data curation, T.C.B.; writing—original draft preparation, T.C.B.; writing—review and editing, T.C.B., D.A.K., and C.E.S.; visualization, T.C.B., D.A.K., and C.E.S.; supervision, T.C.B., D.A.K., and C.E.S.; project administration, T.C.B., D.A.K., and C.E.S.; funding acquisition, D.A.K. and C.E.S.

**Funding:** This work was supported by the University of Tennessee Institute of Agriculture.

**Acknowledgments:** A special thanks to J. Wheeler, K. Cressman and C. Whitlock for technical assistance throughout the project.

**Conflicts of Interest:** The authors declare no conflict of interest.

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
