# Peer review of "Applications of Abscisic Acid and Increasing Concentrations of Calcium Affect the Partitioning of Mineral Nutrients between Tomato Leaf and Fruit Tissue"

_horticulturae, doi:10.3390/horticulturae5030049_

Round 1

Reviewer 1 Report

This manuscript considers the effect of ABA applied as a spray on to the roots, and of calcium concentration, on nutrient concentrations in tomato plants. The authors should give some references to the role of calcium in blossom end rot. The authors should not use consequently, interestingly, furthermore, specifically, etc. Keep sentences short and to the point. Do not state results are significantly different. Only give those results that are significant. When presenting results of other publications, explain how they are different than those given here.

Line 25 Use ‘through’ not ‘by’

Line 34 35 Use ‘The availability and root uptake,’

Line 41 44 Give references here

Line 64 65 Give references here.

Line 75 and elsewhere Use ‘increasing Ca concentrations’

Line 78 Use ‘affecting plant growth by reducing transpiration xylem tissue’

Line 130 131 In what way were the two experiments similar statistically?

Line 140 142 replace with ‘Phosphorus was increased by 16% in tomato leaf tissue when ABA was applied as a foliar spray.’

Line 166 167 replace with ‘The K and P concentrations in the fruit tissue were decreased by 9% and 11%, respectively, compared to the control treatment, when ABA was applied as a foliar spray.’

Line 189 Delete ‘treatment’

Line 225 226 Replace with ‘The Mg, P, and K decreased 30%, 14%, and 11%, respectively, in tomato leaf tissue when Ca treatments increased from 60 to 180 mgL-1, respectively. There was no changes for S in the leaf…’

Lie 249 2450 Change to ‘Calcium did not affect mineral nutrient concentrations in tomato fruit tissue (Table 6).’

Line 270 272 Change to ‘[27], but previous research indicated that ABA indirectly helps to assimilate and distribute Ca into fruit tissue under environmental stress conditions [8-10].’

Line 273 275 Delete the first sentence. Under conditions that favor environmental stress, results varied with how ABA was applied and what mineral nutrient was analyzed.

Line 280 283 change to ‘Application of ABA to tomato plant roots lowered Mg in the leaf tissues more than ABA applied to the foliar tissue. Applying ABA to tomato plant roots may induce a stress response that decreases the uptake and/or distribution of Mg into the leaf tissue’

Line 290 Delete ‘Furthermore, in the current study,’

Line 310 311 What does this sentence refer to?

Line 306 307 change to ‘Because mineral nutrients are taken up by the roots’

Line 329 331 Change to ‘Previous studies showed that ABA inhibits K+ permeability to the xylem and, therefore, reduces its concentrations in leaf and fruit tissue (Give References).’

Line 355 357 change to ‘Kopsell et al. [52] found that increasing Ca concentrations in the nutrition solution decreases Mg, P, K, S, and Zn in kale leaf tissue.’ Delete next sentence.

Line 374 275 change to ‘How Ca influences uptake of mineral nutrients may depend on differences among plant species.’

Author Response

Line 25 Use ‘through’ not ‘by’

On line 25 there is no ‘by’. The authors will omit the suggestion.

Line 34 35 Use ‘The availability and root uptake,’

Thank you for the suggestion. The authors have made taken out ‘consequently’.

Line 41 44 Give references here

Added references 5-8.

Line 64 65 Give references here.

Added reference 19

Line 75 and elsewhere Use ‘increasing Ca concentrations’

Thank you for the suggestion. The author have changed relevant wording for ‘increasing Ca concentrations’.

Line 78 Use ‘affecting plant growth by reducing transpiration xylem tissue’

The authors have added ‘by reducing transpiration’ in line 78.

Line 130 131 In what way were the two experiments similar statistically?

The two experiments were analyzed and there were not interaction between the two experiments. Thus, the data from the two experiments were pooled into one data set.

Line 140 142 replace with ‘Phosphorus was increased by 16% in tomato leaf tissue when ABA was applied as a foliar spray.’

Thank you for the suggestion. The authors have made the wording change.

Line 166 167 replace with ‘The K and P concentrations in the fruit tissue were decreased by 9% and 11%, respectively, compared to the control treatment, when ABA was applied as a foliar spray.’

Thank you for the suggestion. The authors have made the wording changes.

Line 189 Delete ‘treatment’

Deleted ‘treatment’

Line 225 226 Replace with ‘The Mg, P, and K decreased 30%, 14%, and 11%, respectively, in tomato leaf tissue when Ca treatments increased from 60 to 180 mg×L-1, respectively. There was no changes for S in the leaf…’

Thank you for the suggestion. The authors have made the wording changes.

Lie 249 2450 Change to ‘Calcium did not affect mineral nutrient concentrations in tomato fruit tissue (Table 6).’

The authors deleted ‘treatments, in general,’

Line 270 272 Change to ‘[27], but previous research indicated that ABA indirectly helps to assimilate and distribute Ca into fruit tissue under environmental stress conditions [8-10].’

Thank you for the suggestion. However, the authors believe that the current wording if sufficient.

Line 273 275 Delete the first sentence. Under conditions that favor environmental stress, results varied with how ABA was applied and what mineral nutrient was analyzed.

Thank you for the suggestion. The authors have deleted the sentence.

Line 280 283 change to ‘Application of ABA to tomato plant roots lowered Mg in the leaf tissues more than ABA applied to the foliar tissue. Applying ABA to tomato plant roots may induce a stress response that decreases the uptake and/or distribution of Mg into the leaf tissue’

Thank you for the suggestions. The authors have changed the wording to the manuscript.

Line 290 Delete ‘Furthermore, in the current study,’

Deleted

Line 310 311 What does this sentence refer to?

After reading through the paragraph, the authors have deleted the sentence and reference.

Line 306 307 change to ‘Because mineral nutrients are taken up by the roots’

The authors have made the proper changes.

Line 329 331 Change to ‘Previous studies showed that ABA inhibits K+ permeability to the xylem and, therefore, reduces its concentrations in leaf and fruit tissue (Give References).’

Thank you for your suggestion. However, the authors believe that the current wording of the two sentences explains the context of the subject matter. The second sentence is a continuation of ref [45] from the previous sentence, just further explaining the relationship.

Line 355 357 change to ‘Kopsell et al. [52] found that increasing Ca concentrations in the nutrition solution decreases Mg, P, K, S, and Zn in kale leaf tissue.’ Delete next sentence.

The authors have deleted the second sentence.

Line 374 275 change to ‘How Ca influences uptake of mineral nutrients may depend on differences among plant species.’

Thank you for the suggestion. We have deleted the first part of the sentence and added How.

Reviewer 2 Report

Review Report Abstract a) Line 17: ... with different concentrations of ABA and Ca. b) Lines 22-23: The results section in the abstract should describe how ABA applied to the leaf or root affect nutrient concentrations in leaf and fruit. Material and Methods a) Line 116: Sampling dates for both fruit and leaf must be precisely described in the methods. Different sampling times have different responses. In addition, it is required to describe which leaf and fruit were used to each analysis. b) Calcium and ABA applications through the roots must be better described. The application time and frequency must be fully described. Results a) All Tables must describe the fruit, leaf and sampling time used in the study. Fruit and leaf ages, as well as sampling time are essential to understand the results. b) Lines 136 and 142: Sampling times must be stated in the text. c) I do not understand the reason why the manuscript does not present the concentrations of nitrogen and calcium in leaf and fruit tissues. The manuscript must present these results to fully show the effect of the treatment on the most important nutrients. d) All tables must have similar detailed description of the experiment. Please, revise all table titles. Ex: Table 1 and Table 2 are very different. Discussion I suggest adding a diagram presenting the possible mechanisms through which ABA and Ca are regulating leaf and fruit mineral composition. This diagram must summarize all results obtained in the study.

Author Response

Reviewer 2: Review Report

Abstract

a)     Line 17: ... with different concentrations of ABA and Ca.

Added ‘of’

b) Lines 22-23: The results section in the abstract should describe how ABA applied to the leaf or root affect nutrient concentrations in leaf and fruit.

Thank you for the insight. However, the abstract is limited to 200 word and the authors can not find a way to describe the varying effects of ABA applications to mineral nutrient concentrations in such a short abstract. Thus, a general statement has to be made in the abstract in order to meet the word restriction. Some of the minerals were increased and some decreased with ABA applications and some mineral nutrients were specific in the way ABA was applied (leaf tissue and/or root tissue).

Material and Methods a) Line 116: Sampling dates for both fruit and leaf must be precisely described in the methods. Different sampling times have different responses. In addition, it is required to describe which leaf and fruit were used to each analysis. b) Calcium and ABA applications through the roots must be better described. The application time and frequency must be fully described.

Thank you for your suggestions. The authors were asked by Horticulturae Editors to cut down on the material and methods section. Thus, we cut out some of the descriptive details of the study. A full materials and methods section can be found in REF [10]. There were multiple dates of tomato fruit harvest because once the fruit were red ripe on the cluster, the fruit were harvested. Multiple fruit cluster were harvested throughout the experiment. Thus, a general time frame was used to describe the fruit harvest in this manuscript. Additionally, the times and frequency were also cut from this manuscript but described in greater details in REF [10].

The authors did add that the fruit and leaf tissue taken from the plant were harvested at 84 days after seeding from the second cluster for mineral nutrient analysis.

Results

a)     All Tables must describe the fruit, leaf and sampling time used in the study. Fruit and leaf ages, as well as sampling time are essential to understand the results.

The authors added to the tables that the fruit or leaves were harvested 84 d after seeding for the mineral nutrient analysis.

b)    Lines 136 and 142: Sampling times must be stated in the text.

The sampling time was described in the materials and methods section.

c)     I do not understand the reason why the manuscript does not present the concentrations of nitrogen and calcium in leaf and fruit tissues. The manuscript must present these results to fully show the effect of the treatment on the most important nutrients.

Nitrogen was not analyzed because we did not have the capability to measure N in our lab at the time that the experiments were conducted. Data for the calcium was published in

[10] Barickman, T.C., D.A. Kopsell, and C.E. Sams. 2014c. Exogenous foliar and root applications of abscisic acid increase the influx of calcium into tomato fruit tissue and decrease the incidence of blossom. HortScience 49: 1397-1402.

d)    All tables must have similar detailed description of the experiment. Please, revise all table titles. Ex: Table 1 and Table 2 are very different. Discussion I suggest adding a diagram presenting the possible mechanisms through which ABA and Ca are regulating leaf and fruit mineral composition. This diagram must summarize all results obtained in the study.

Thank you for your suggestion. We have added more description to the tables to help in the understanding of the data. However, the authors do not feel that another diagram is necessary to help in understanding the relationship between ABA, Ca, and mineral nutrients composition for this manuscript due to the variability of how ABA (and how it was applied) affected the leaf and fruit tissue concentrations.